# Patient-level interventions to reduce alcohol-related harms in low- and middle-income countries: A systematic review and meta-summary

Catherine A. Staton[1,2,3]*, João Ricardo Nickenig Vissoci[1,2,3], Deena El-Gabri[2], Konyinsope Adewumi[2], Tessa Concepcion[2], Shannon A. Elliott[2], Daniel R. Evans[2], Sophie W. Galson[1,2], Charles T. Pate[2], Lindy M. Reynolds[2], Nadine A. Sanchez[2], Alexandra E. Sutton[2,4], Charlotte Yuan[2], Alena Pauley[2], Luciano Andrade[3], Megan Von Isenberg[5], Jinny J. Ye[1], Charles J. Gerardo[1,2]

1 Duke Division of Emergency Medicine, Department of Surgery, Duke University Medical Center, Duke University, Durham, North Carolina, United States of America, 2 Duke Global Health Institute, Duke University, Durham, North Carolina, United States of America, 3 Health Sciences Graduate Program, State University of Maringa, Maringa, Parana State, Brazil, 4 Nicholas School of the Environment, Duke University, Durham, North Carolina, United States of America, 5 Duke School of Medical Center Library Services & Archives, Duke University, Durham, North Carolina, United States of America

* Catherine.staton@duke.edu

**Data Availability Statement:** All related data and metadata for the reported findings are available at doi.org/10.6084/m9.figshare.13836668.

## Abstract

### Background

Disease and disability from alcohol use disproportionately impact people in low- and middle-income countries (LMICs). While varied interventions have been shown to reduce alcohol use in high-income countries, their efficacy in LMICs has not been assessed. This systematic review describes current published literature on patient-level alcohol interventions in LMICs and specifically describes clinical trials evaluating interventions to reduce alcohol use in LMICs.

### Methods and findings

In accordance with PRISMA, we performed a systematic review using an electronic search strategy from January 1, 1995 to December 1, 2020. Title, abstract, as well as full-text screening and extraction were performed in duplicate. A meta-summary was performed on randomized controlled trials (RCTs) that evaluated alcohol-related outcomes. We searched the following electronic databases: PubMed, EMBASE, Scopus, Web of Science, Cochrane, WHO Global Health Library, and PsycINFO. Articles that evaluated patient-level interventions targeting alcohol use and alcohol-related harm in LMICs were eligible for inclusion. No studies were excluded based on language.

After screening 5,036 articles, 117 articles fit our inclusion criteria, 75 of which were RCTs. Of these RCTs, 93% were performed in 13 middle-income countries, while 7% were from 2 low-income countries. These RCTs evaluated brief interventions (24, defined as any intervention ranging from advice to counseling, lasting less than 1 hour per session up to 4 sessions), psychotherapy or counseling (15, defined as an interaction with a counselor longer than a brief intervention or that included a psychotherapeutic component), health

**Funding:** CAS received salary support funding from the Fogarty International Center under Staton, K01 TW010000-01A1 (URL: https://www.fic.nih.gov/). The funders had no role in study design, data collection and analysis, decision to publish, or preparation of the manuscript.

**Competing interests:** The authors have declared that no competing interests exist.

**Abbreviations:** AA, Alcoholics Anonymous; ACROBAT-NRS, A Cochrane Risk Of Bias Assessment Tool for Non-Randomized Studies; ASSIST, Alcohol, Smoking and Substance Involvement Screening Test; AUD, alcohol use disorder; AUDIT, Alcohol Use Disorders Identification Test; CBT, cognitive behavioral therapy; DALY, disability-adjusted life year; LMIC, low- and middle-income country; MI, motivational interviewing; NOS, Newcastle–Ottawa scale; PNF, personalized normative feedback; PRISMA, Preferred Reporting Items for Systematic Reviews and Meta-Analyses; RAPI, Rutgers Alcohol Problem Index; RCT, randomized controlled trial; SADQ, Severity of Alcohol Dependence Questionnaire; STEP, School-based Teenage Education Program; STROBE, STrengthening the Reporting of OBservational studies in Epidemiology; WHO, World Health Organization.

promotion and education (20, defined as an intervention encouraged individuals' agency of taking care of their health), or biologic treatments (19, defined as interventions where the biological function of alcohol use disorder (AUD) as the main nexus of intervention) with 3 mixing categories of intervention types. Due to high heterogeneity of intervention types, outcome measures, and follow-up times, we did not conduct meta-analysis to compare and contrast studies, but created a meta-summary of all 75 RCT studies. The most commonly evaluated intervention with the most consistent positive effect was a brief intervention; similarly, motivational interviewing (MI) techniques were most commonly utilized among the diverse array of interventions evaluated.

## Conclusions

Our review demonstrated numerous patient-level interventions that have the potential to be effective in LMICs, but further research to standardize interventions, populations, and outcome measures is necessary to accurately assess their effectiveness. Brief interventions and MI techniques were the most commonly evaluated and had the most consistent positive effect on alcohol-related outcomes.

## Trial registration

Protocol Registry: PROSPERO CRD42017055549

## Author summary

### Why was this study done?

- Low- and middle-income countries (LMICs) report high rates of risky alcohol use behavior, a known risk factor for death and disability worldwide.

- In order to investigate the potential for a patient-level intervention to reduce alcohol-related harms in a low-income setting, we sought to identify interventions with adequate efficacy.

### What did the researchers do and find?

- We conducted a systematic review of studies from 1995 to 2020 in LMICs evaluating interventions to reduce alcohol use and alcohol-related harms.

- Of the 117 studies included for review, the majority were in middle-income countries and had varied intervention types, outcome measures, and follow-up time.

- The most commonly studied interventions with the most consistently positive results were brief interventions. Similarly, motivational interviewing (MI) techniques were the most commonly described intervention techniques.

### What do these findings mean?

- Future research on alcohol use and alcohol harm reduction in LMICs may benefit from consistency of methodologies, studying similar populations, interventions, and alcohol-related harm reduction outcome measures.

- Especially in LMICs, further research on comparative effectiveness or implementation strategies delineating optimal interventions and target populations is needed.

## Introduction

Alcohol use is an important cause of chronic disease and injury. It is one of the top 5 risk factors for death and disability in the world [1–3]. The detrimental effects of alcohol use contribute to 3.3 million deaths and 139 million disability-adjusted life years (DALYs) lost globally each year [4]. Alcohol use has also been associated with risky behaviors, including crime, aggressive driving, interpersonal violence, and self-inflicted injury [5]. Such behaviors not only have harmful effects on the individual but also on the greater population [6]. Compared to high-income countries, low- and middle-income countries (LMICs) report higher rates of risky drinking behaviors, such as binge drinking and episodic drinking, as well as an earlier onset of alcohol consumption [4].

The World Health Organization (WHO) has placed an emphasis on the development and implementation of both policy-level and patient-level interventions to reduce harmful alcohol use in LMICs. While policy-level interventions are a crucial, cost-effective manner of reducing alcohol-related harms, context-appropriate, and effective patient-level interventions are also greatly needed to form multipronged alcohol harm reduction strategies [4]. A broad array of patient-level alcohol harm reduction interventions, such as brief interventions (for this paper, defined as any intervention ranging from advice to counseling, lasting less than 1 hour per session [7] up to 4 sessions [8], psychosocial interventions, and pharmacological treatments) have been found to be effective in high-resource settings [9,10]. Yet, alcohol use disorders (AUDs), characterized by moderate to severe alcohol abuse and dependence, remain a low priority of LMIC health systems [11]. Barriers, such as funding constraints, lack of policy, and low public awareness, often prevent access to psychosocial and pharmacological treatments that target AUDs [11]. Especially in some settings where alcohol use is culturally ingrained, adopting an alcohol harm reduction strategy, as opposed to focusing on abstinence, is crucial given the limited alcohol policy, health system treatments, and social support [12]. As such, WHO and *The Lancet* have recently issued calls to action to reduce hazardous alcohol use [4,13], yet the full scope of the evidence-based patient-level interventions to reduce harmful alcohol use in LMICs is missing from the literature. While narrative reviews of global alcohol-related harms have been published, we have found no systematic review conducted focusing on alcohol interventions specifically applicable to or evaluated in LMICs [1,11].

In order to address this gap, this paper aims to (1) review and describe the current published literature on patient-level alcohol interventions in LMICs; and (2) conduct a meta-summary of studies evaluating interventions to reduce alcohol use and harms in LMICs.

## Methods

### Protocol and registration

This systematic review is reported in accordance with the Preferred Reporting Items for Systematic Reviews and Meta-Analyses (PRISMA) Statement [14] (see S1 Table) and is registered in the PROSPERO database (International Prospective Register of Systematic Reviews) under the number CRD42017055549.

## Eligibility criteria

Our primary criterion for article consideration was a patient-level alcohol or alcohol-related harm reduction intervention in a LMIC, as defined by our PICOS framework: LMIC **P**artici-pants, patient-level **I**nterventions, **C**ompared to a control group, alcohol harm reduction **O**ut-comes, all **S**tudy designs but focused on randomized clinical trials if there are enough. To be included, articles had to (1) evaluate a patient-level alcohol-related intervention's ability to reduce an (2) alcohol-related outcome in a (3) LMIC and be (4) peer-reviewed and published between January 1, 1995 and December 1, 2020. Study locations had to be classified as LMICs according to World Bank criteria at the time of the search [15]. The search strategy was inclusive of multiple study designs (randomized controlled trials [RCTs], prospective/retrospective cohort, quasi-experimental, or secondary data analyses with before and after intervention comparison) in case there was a dearth of literature from LMIC settings. Articles were excluded if they were abstracts only, literature or systematic reviews, meta-analyses, or commentaries. If 2 studies used the same data, then the most recent data were included in the review.

## Information sources

We searched electronic databases (PubMed, EMBASE, Scopus, Web of Science, Cochrane, WHO Global Health Library, and PsycINFO) for articles that evaluated patient-level interventions aimed at reducing an alcohol-related outcome in LMICs. No studies were excluded for language. Additionally, we manually searched references and performed a citation analysis of the included articles using Web of Science and Google Scholar. Any citation that met the inclusion criteria based on the title and abstract was added.

## Search

The initial search consisted of the MeSH terms "alcohol drinking," "low or middle income country," and "intervention." Search strategy demonstrates the search strategy used in PubMed, Embase, PsycINFO, and WHO Global Health Library databases (S1 Fig).

## Study selection

Six pairs of reviewers from the specified individuals (KA, TC, SE, DE, SG, CP, LR, NS, AS, CY, and AP) independently reviewed the titles and abstracts, and any inconsistencies regarding inclusion were resolved by a third reviewer (DG or CS). Abstracts that did not provide enough information to determine eligibility were retrieved for full-text evaluation. Reviewers independently evaluated full-text articles and determined study eligibility. Disagreements were solved by consensus, and if disagreement persisted, a third reviewer's opinion was sought. After inclusion, we assessed each study for the study design. We reported all study designs in order to summarize the type and quality of study designs in the literature. Based on the large number of RCTs identified, we chose to narrow further analysis to RCTs.

## Quality of studies

Since our systematic review included studies of different designs (RCTs, nonrandomized intervention, prospective/retrospective cohort, quasi-experimental, or secondary data/cross-sectional with before and after comparison), we opted to perform a data quality assessment according to study design using the following approaches. STrengthening the Reporting of OBservational studies in Epidemiology (STROBE) indicators were used for reporting observational studies. Two scales were used for nonrandomized studies: the A Cochrane Risk Of Bias

Assessment Tool for Non-Randomized Studies (ACROBAT-NRS) [16] and Newcastle–Ottawa scale (NOS) [17]. Cochrane's revised risk-of-bias tool was used for randomized studies [18]. Finally, the Effective Practice and Organisation of Care (EPOC) suggested risk of bias indicators for interrupted time series studies (EPOC) [19]. We assigned risk of bias (low, moderate, and high risk) as suggested by the Cochrane Handbook [20] by study design. Studies were classified as (a) low risk of bias if all domains had low risk; (b) some concerns if at least 1 domain raised some concerns for bias; and (c) high risk of at least 1 domain was at high risk.

## Data extraction

Five pairs of reviewers independently conducted the data extraction, and any disagreements were resolved by a third reviewer. General characteristics of the studies were recorded, such as year of publication, location where the study took place, inclusion and exclusion criteria, and participant characteristics. In addition, information on alcohol-related outcome measures, intervention type, and intervention impact or effectiveness measured as an effect size of outcome measures was extracted. The main outcome measures were Alcohol Use Disorders Identification Test (AUDIT) and Alcohol, Smoking and Substance Involvement Screening Test (ASSIST) scores, Rutgers Alcohol Problem Index (RAPI), number of drinking days, number of heavy drinking days, number of binge drinking days, drinks per drinking day, percent remaining abstinent from drinking alcohol, and percent relapsed back into drinking alcohol.

## Data analysis

Initial evaluation of the papers indicated that a meta-analytical approach would result in high heterogeneity due to high methodological variability (e.g., outcome measures, study designs, and sample characteristics). Therefore, we conducted a meta-synthesis for all the included manuscripts, which qualitatively aggregated findings by grouping relevant findings into categories that represent the study's objectives (e.g., effectiveness of alcohol intervention). No manuscripts were excluded based on quality. The process involved summarizing main results of each included paper and performing a thematic analysis. Emerging themes on types of intervention and outcomes were presented. Interventions were grouped by similarity into 4 types: brief interventions, psychotherapy and counseling, health promotion and education, and biomedical treatments.

Using WHO and National Institute on Alcohol Abuse and Alcoholism (NIAAA) descriptions of brief interventions, we defined brief interventions as any intervention ranging from advice to counseling, lasting less than 1 hour per session [7] up to 4 sessions [8] independent of how the original study defined brief intervention. Interventions including a one-on-one interaction with a counselor that lasted longer than a brief intervention or that included a psychotherapeutic component were defined as psychotherapy and counseling. Motivational interviewing (MI) techniques could be included as either a brief intervention or psychotherapy and counseling, depending on how long and over how many sessions the intervention took place. A study was considered health promotion and education, independent of the study's definition, if an intervention encouraged individuals' agency of taking care of their health, such as risk reduction skills and health education [21]. Biomedical treatments were used as a taxonomy to group studies that had the biological function of AUD as the main nexus of intervention, including brain stimulation and medicines.

## Results

### Study selection and description

In total, 5,036 abstracts were reviewed. From those, 500 articles were manually reviewed to identify 117 articles matching our inclusion and exclusion criteria (Fig 1). No studies were

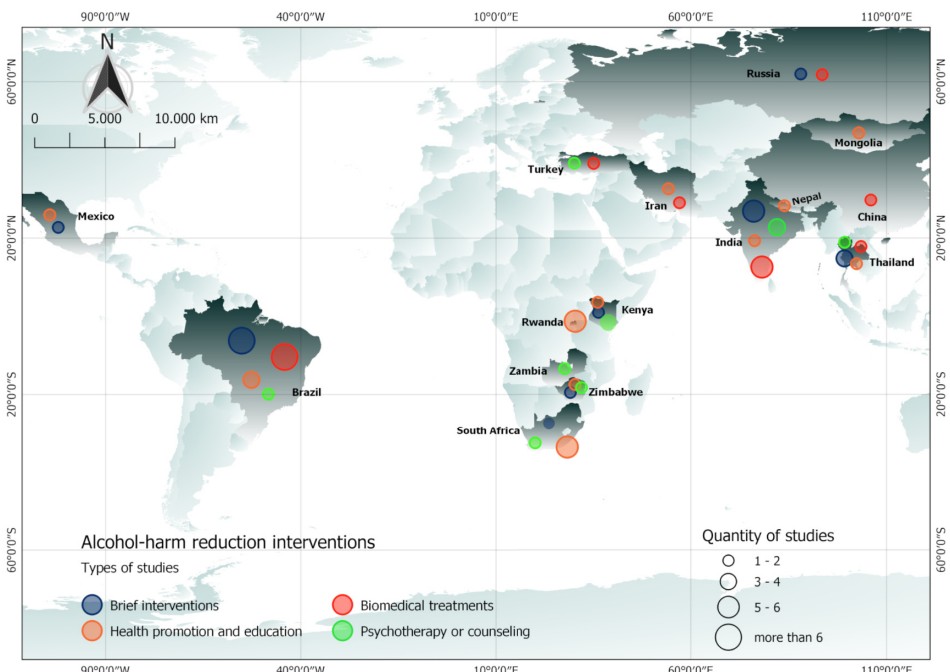

**Fig 1. Study flow diagram.**

excluded based on language. Of these 117 studies, 75 were RCTs (Table 1) utilizing a vast array of interventions, which we categorized into 4 main categories of interventions, including brief interventions (24 studies) (Table 2), psychotherapy or counseling (15) (Table 3), health promotion and education (20) (Table 4), and biological treatments (19) (Table 5). One study by Shin and colleagues had one arm in biomedical treatments and another arm in brief intervention [22]. Two other studies had one arm in psychotherapy or counseling and another arm in health promotion and education [23,24]. These 75 studies were performed in 15 countries, representing 8 upper middle-income countries (60% of studies), 5 lower middle-income countries, and 2 low-income countries (7% of studies) (S2 Fig). The majority of the studies came from Brazil (28%) and India (20%). Alcohol-related outcomes found included alcohol quantity or frequency measure, intention to use alcohol, use/abstinence/remission proportion or frequency, alcohol-related scores, alcohol cravings or cravings per day, or alcohol use during pregnancy or before sex.

## Meta-summary

**Brief interventions.**   The brief interventions category had the greatest number of RCTs in our study with 24 RCTs, and these interventions were the most similar to each other. The types of interventions included most commonly were WHO-based brief interventions (which utilizes some MI techniques) [28,29,79–82] or MI interventions [22,58,76,88,89,91,93]. Some studies focused more on the intervention delivery, specifically nurse or layperson [67,70,71,75] or computer-based interventions [30,34,35,41]. Outcomes were also varied including harmful alcohol use scores (AUDIT) or alcohol misuse (ASSIST), abstinence or remission (ASSIST), and percent or number of days of drinking or heavy drinking.

Overall, the majority of the studies evaluating brief interventions demonstrated evidence of efficacy in one or more of their alcohol-related outcomes, for both short- (up to 3 months) and

**Table 1. Characteristics of all randomized controlled studies (75).**

| Authors | Country | Intervention type | Targeted population | Sample size | Risk of bias | Outcomes measured |
|---|---|---|---|---|---|---|
| Ahmadi and colleagues (2004) [24] | Iran | Biomedical treatments | Self-referred, alcohol-dependent males | 116 | High | Relapse |
| Ahmadi and colleagues (2019) [25] | Iran | Health promotion and education | Female drug users | 100 | High | Alcohol use before sexual intercourse |
| Aira and colleagues (2013) [26] | Mongolia | Health promotion and education | Power plant employees | 200 | Low | Drinking days Drinks per day |
| Altintoprak and colleagues (2008) [27] | Turkey | Biomedical treatments | AUD patient population | 44 | Low | Alcohol use Craving |
| Assanangkornchai and colleagues (2015) [28] | Thailand | Brief intervention | Primary care | 236 | Low | ASSIST Conversion to low risk |
| Babor and colleagues (1996) [29] | Australia, Kenya, Mexico, Norway, Wales, Russia, USA, and Zimbabwe | Brief intervention | Users at risk for dependence in hospital, emergency department, primary care, college, and health screening agency | 1,559 | High | Abstinence Frequency Intensity Harm (injury, legal problem, and unemployment) Complaint from others |
| Baldin and colleagues (2018) [30] | Brazil | Brief intervention | Nightclub users with drinking problems | 465 | Low | Binge drinking Lack of control |
| Baltieri and colleagues (2003) [31] | Brazil | Biomedical treatments | Alcohol-dependent males in outpatient treatment | 75 | Low | Abstinence |
| Baltieri and colleagues (2008) [32] | Brazil | Biomedical treatments | Alcohol-dependent males in outpatient treatment | 155 | High | Abstinence/relapse Weeks of heavy consumption |
| Barbosa Filho and colleagues (2019) [33] | Brazil | Health promotion and education | School-based adolescents | 1,085 | Low | Alcohol intake |
| Bedendo and colleagues (2019) [34] | Brazil | Brief intervention | College drinkers | 4,460 | Some concerns | AUDIT Alcohol-related consequences Drinks per drinking day |
| Bedendo and colleagues (2019) [35] | Brazil | Brief Intervention | College drinkers | 5,476 | Some concerns | AUDIT Alcohol-related consequences Drinking days Drinks per drinking days |
| Boggio and colleagues (2008) [36] | Brazil | Biomedical treatments | Alcohol-dependent users in rehabilitation program | 13 | Low | Alcohol Urge Questionnaire (craving level) |
| Bolton and colleagues (2014) [37] | Thailand | Health promotion and education | Survivors of imprisonment, torture, and related traumas | 347 | Low | Alcohol use |
| Burnhams and colleagues (2015) [38] | South Africa | Health promotion and education | Safety and security employees | 325 | Low | Binge drinking days Calling in sick or working with a hangover CAGE |
| Chaudhury and colleagues (2016) [39] | Rwanda | Health promotion and education | Families with caregiver HIV | 293 | Low | AUDIT |
| Chhabra and colleagues (2010) [40] | India | Health promotion and education | Teenage students | 1,421 | Low | Future intentions to use |
| Christoff and colleagues (2015) [41] | Brazil | Brief intervention | College students | 815 | Some concerns | ASSIST |
| Corrêa Filho and colleagues (2013) [42] | Brazil | Biomedical treatments | Alcohol-dependent males in outpatient treatment | 102 | Low | Drinks per day Abstinence Heavy drinking days |
| Cubbins and colleagues (2012) [43] | Zimbabwe | Health promotion and education | Rural communities | 5,543 | High | Abstinence Drinks per drinking day Drinking days Drunk days |
| da Silva and colleagues (2013) [44] | Brazil | Biomedical treatments | Alcohol-dependent users in outpatient treatment | 13 | Low | Relapse OCDS Alcohol Urge Questionnaire |

*(Continued)*

**Table 1.** (Continued)

| Authors | Country | Intervention type | Targeted population | Sample size | Risk of bias | Outcomes measured |
|---|---|---|---|---|---|---|
| Daengthoen and colleagues (2014) [45] | Thailand | Psychotherapy or counseling | Alcohol-dependent users in inpatient treatment | 100 | Low | Craving days<br>Abstinent days<br>Drinking days |
| De Sousa and colleagues (2004) [46] | India | Biomedical treatments | Private hospital adult psychiatric patients | 100 | High | Abstinence days<br>Days until relapse<br>Drinks per drinking day<br>Craving |
| De Sousa and colleagues (2005) [47] | India | Biomedical treatments | Private hospital adult psychiatric patients | 100 | High | Abstinence days<br>Days until relapse<br>Drinks per drinking day<br>Craving |
| De Sousa and colleagues (2008) [48] | India | Biomedical treatments | Private hospital adolescent psychiatric patients | 100 | High | Abstinence days<br>Days until relapse<br>Drinks per drinking day<br>Craving |
| De Sousa and colleagues (2008) [49] | India | Biomedical treatments | Private hospital adult psychiatric patients | 100 | High | Abstinence days<br>Days until relapse<br>Drinks per drinking day<br>Craving |
| De Sousa and colleagues (2014) [50] | India | Biomedical treatments | Private hospital adult psychiatric patients | 100 | High | Abstinence days<br>Days until relapse<br>Drinks per drinking day<br>Craving |
| Furieri and colleagues (2007) [51] | Brazil | Biomedical treatments | Alcohol-dependent users referred for alcohol treatment | 60 | Low | Drinks per day<br>Drinks per drinking day<br>Heavy drinking days<br>Percent abstinent<br>OCDS |
| Gupta and colleagues (2017) [52] | India | Biomedical treatments | Alcohol-dependent users in outpatient treatment | 122 | Some concerns | Heavy drinking days<br>Abstinent days<br>Days to first relapse<br>Relapse<br>Abstinence<br>OCDS |
| Hartmann and colleagues (2020) [53] | India | Psychotherapy or counseling | Couples | 60 couples | Some concerns | Alcohol Breathalyzer<br>Abstinence |
| Jirapramukpitak and colleagues (2020) [54] | Thailand | Health promotion and education | Alcohol-dependent users | 161 | Some concerns | Abstinence |
| Jordans and colleagues (2019) [55] | Nepal | Health promotion and education | Mental health patients at the primary care setting | 162 | Low | AUDIT |
| Kalichman and colleagues (2008) [56] | South Africa | Health promotion and education | Users at informal drinking establishment | 353 | Low | Alcohol outcome expectancy ("I am a better sex partner after I have been drinking" and "When I'm drinking, I do things I wouldn't usually do") |
| Kalichman and colleagues (2007) [57] | South Africa | Health promotion and education | Sexually transmitted infections clinic | 143 | Low | |
| Kamal and colleagues (2020) [58] | India | Brief Intervention | College students with hazardous use | 130 | Low | AUDIT |
| Klauss and colleagues (2014) [59] | Brazil | Biomedical treatments | Alcohol-dependent users | 33 | Low | Relapse<br>OCDS |
| Klauss and colleagues (2018) [60] | Brazil | Biomedical treatments | Alcohol-dependent users | 45 | Low | Relapse<br>OCDS |
| L'Engle and colleagues (2014) [61] | Kenya | Psychotherapy or counseling | Female sex workers | 818 | High | Drinks per week<br>Binge drinking |

(*Continued*)

**Table 1.** (Continued)

| Authors | Country | Intervention type | Targeted population | Sample size | Risk of bias | Outcomes measured |
|---|---|---|---|---|---|---|
| Likhitsathian and colleagues (2013) [62] | Thailand | Biomedical treatments | Inpatient treatment for AUDs | 106 | Low | Heavy drinking days<br>Drinks per day<br>Drinks per drinking day<br>Cravings (visual analog)<br>Heavy drinking relapses |
| Madhombiro and colleagues (2020) [63] | Zimbabwe | Psychotherapy or counseling | HIV clinic | 234 | Low | AUDIT |
| Marques and colleagues (2001) [64] | Brazil | Psychotherapy or counseling | Alcohol-dependent users | 155 | High | Drinking days<br>Heavy drinking days<br>Problem drinking days<br>Drinks per week<br>Alcohol Dependence Data Questionnaire |
| Marsiglia and colleagues (2015) [65] | Mexico | Health promotion and education | Middle school students | 431 | Some concerns | Drinks per month<br>Drinking days |
| Mendez-Ruiz and colleagues (2020) [66] | Mexico | Health promotion and education | Sexually active female college students | 132 | Some concerns | AUDIT |
| Mertens and colleagues (2014) [67] | South Africa | Brief intervention | Young adults from primary care clinic | 403 | Low | ASSIST<br>Heavy drinking |
| Moraes and colleagues (2010) [23] | Brazil | Psychotherapy or counseling and Health promotion and education | Alcohol-dependent users in outpatient treatment | 120 | High | Abstinence<br>Drinking days<br>Addiction Severity Index |
| Murray and colleagues (2020) [68] | Zambia | Psychotherapy or counseling | Couples with male hazardous use and intimate partner violence | 248 couples | Low | AUDIT |
| Nadkarni and colleagues (2015) [69] | India | Brief intervention | Males presenting to primary care | 53 | Low | AUDIT |
| Nadkarni and colleagues (2017) [70] | India | Brief intervention | Harmful drinking in males in primary care | 377 | Low | AUDIT<br>Abstinence<br>Amount of consumption<br>Heavy drinking days |
| Nadkarni and colleagues (2017) [71] | India | Brief Intervention | Harmful drinking in males in primary care | 377 | Low | AUDIT<br>Abstinence<br>Amount of consumption |
| Nadkarni and colleagues (2019) [72] | India | Brief Intervention | Alcohol-dependent males | 135 | Low | Remission (AUDIT <8)<br>Mean daily alcohol consumption<br>% abstinent<br>% heavy drinking days<br>Uptake of detoxification services<br>SIP |
| Nattala and colleagues (2010) [73] | India | Psychotherapy or counseling | Inpatient | 90 | High | Abstinence<br>Amount of consumption<br>Drinking days |
| Ng and colleagues (2020) [74] | India | Psychotherapy or counseling | Alcohol-dependent users | 60 | Some concerns | Craving<br>Drinking days<br>Drinks per drinking days<br>Relapse |
| Noknoy and colleagues (2010) [75] | Thailand | Brief intervention | Harmful users in primary care | 117 | Low | Drinks per drinking day<br>Hazardous drinking<br>Drinks per week<br>Binge drinking |
| Pal and colleagues (2007) [76] | India | Brief intervention | Male harmful users | 90 | Some concerns | Drinking days<br>Addiction Severity Index |

(*Continued*)

**Table 1.** (Continued)

| Authors | Country | Intervention type | Targeted population | Sample size | Risk of bias | Outcomes measured |
|---|---|---|---|---|---|---|
| Papas and colleagues (2011) [77] | Kenya | Psychotherapy or counseling | HIV clinic | 75 | Low | Drinking days<br>Drinks per drinking day<br>Abstinence |
| Papas and colleagues (2020) [78] | Kenya | Psychotherapy or counseling and Health promotion and education | HIV clinic | 614 | Low | % drinking days<br>Drinks per drinking day |
| Peltzer and colleagues (2013) [79] | South Africa | Brief intervention | TB patients | 853 | Low | AUDIT<br>Heavy episodic drinking |
| Pengpid and colleagues (2013a) [80] | South Africa | Brief intervention | Hazardous or harmful users | 392 | Some concerns | AUDIT<br>Heavy episodic drinking |
| Pengpid and colleagues (2013b) [81] | South Africa | Brief intervention | University students | 152 | Some concerns | |
| Pengpid and colleagues (2015) [82] | Thailand | Brief intervention | Outpatient clinic | 620 | Low | Drinks per week<br>ASSIST |
| Rendall-Mkosi and colleagues (2013) [83] | South Africa | Psychotherapy and counseling | Pregnant women | 165 | Some concerns | Risky drinking<br>AUDIT |
| Rotheram-Borus and colleagues (2015) [84] | South Africa | Health promotion and education | Pregnant women | 904 | Some concerns | Drinking days<br>Drinks per drinking day<br>Heavy drinking days |
| Sanchez and colleagues (2017) [85] | Brazil | Health promotion and education | Early adolescents (seventh and eighth grades) | 5,028 | Low | Binge drinking episodes |
| Sanchez and colleagues (2018) [86] | Brazil | Health promotion and education | Early adolescents (seventh and eighth grades) | 5,028 | Low | Binge drinking episodes |
| Satyanarayana and colleagues (2016) [87] | India | Psychotherapy and counseling | Alcohol-dependent males in inpatient treatment | 177 | Low | SADQ |
| Segatto and colleagues (2011) [88] | Brazil | Brief intervention | Young adults presenting to emergency department | 175 | Low | Drinking days<br>Light, moderate, or heavy drinking days<br>RAPI<br>ACRQ |
| Shin and colleagues (2013) [22] | Russia | Biomedical treatments and Brief intervention | Adults hospitalized for TB | 196 | Some concerns | Abstinent days<br>Heavy drinking days |
| Signor and colleagues (2013) [89] | Brazil | Brief intervention | Callers to counseling hotline | 637 | High | % abstinent |
| Simao and colleagues (2008) [90] | Brazil | Brief intervention | University students | 266 | Some concerns | RAPI<br>AUDIT<br>Brief Drinker Profile<br>Alcohol Dependence Scale |
| Soares and Vargas (2019) [91] | Brazil | Psychotherapy and counseling | Harmful or hazardous users | 180 | High | AUDIT |
| Sorsdahl and colleagues (2015) [92] | South Africa | Psychotherapy and counseling | Emergency department | 335 | Low | ASSIST |
| Ward and colleagues (2015) [93] | South Africa | Brief intervention | Young adults in primary care | 363 | Low | ASSIST |
| Wechsberg and colleagues (2019) [94] | South Africa | Brief intervention | Black African women going through HIV prevention | 641 | Low | Frequency of heavy drinking episode |
| Witte and colleagues (2011) [95] | Mongolia | Health promotion and education | Female sex workers | 166 | Low | AUDIT |
| Zhao and colleagues (2020) [96] | China | Biomedical treatments | Alcohol-dependent males with withdrawal symptoms | 62 | Some concerns | Craving (PACS) |

ACRQ, Alcohol Consumption Risk Questionnaire; ASSIST, Alcohol, Smoking and Substance Involvement Screening Test; AUD, alcohol use disorder; AUDIT, Alcohol Use Disorders Identification Test; BrAC, breath alcohol content; CAGE, Cut, Annoyed, Guilty, and Eye questionnaire; OCDS, Obsessive Compulsive Drinking Scale; PACS, Penn Alcohol Craving Scale; SADQ, Severity of Alcohol Dependence Questionnaire; SIP, Short Inventory of Problems; TB, tuberculosis.

**Table 2. Meta-synthesis of studies assessing patient-level interventions to reduce alcohol harms in LMICs: brief intervention RCTs.**

| Intervention description | Follow-up time | Outcomes | Summary of findings |
|---|---|---|---|
| WHO-based brief interventions (which uses MI techniques) (6) | 3, 6, 9, and 12 months | Harmful alcohol use score (AUDIT) Alcohol misuses (ASSIST) Heavy drinking % of remission (ASSIST) % abstinent % daily or almost daily drinking % above recommended weekly limit % hazardous average daily consumption | At 3 and 6 months, Assanangkornchai and colleagues found similar significant reductions in the frequency of alcohol use and other substances in both the intervention and control groups at the primary care setting [28]. Penpgid and colleagues did not find evidence of efficacy of a mixed alcohol and tobacco brief intervention compared to an alcohol-only or tobacco-only session on past week alcohol use and Alcohol ASSIST score. All 3 arms did have a significant reduction in their alcohol consumption compared to baseline [82]. Babor and colleagues found at 9 months that males who received brief or simple advice reported a 17% lower average daily alcohol consumption compared to the control group, with a reduction in the intensity of drinking at about 10%. Females reduced their consumption in both groups without between-group differences [29]. Pengpid and colleagues found that at 12 months postintervention, university students had a significant reduction in AUDIT score compared to control [81]. As for outpatients, they found no significant differences in the reduction in relation to control [80]. Peltzer and colleagues evaluated the impact of a brief intervention versus a health leaflet for TB clinic patients and did not find evidence of efficacy between control and intervention at 6 months (79). |
| Face to face and computer based (1) | 3 months | ASSIST | At 3 months, a face-to-face and computer-based MI both reduced ASSIST scores compared to the control group with computer-based intervention with the greater reduction [41]. |
| Nurse, nurse practitioner, or lay counselor delivered (5) | 6 weeks 3, 6, and 12 months | ASSIST % heavy drinking % at-risk use Binge drinking days Drinking days Drinks per day Heavy drinking | Mertens and colleagues found those who received a nurse practitioner–delivered brief intervention reduced patients' alcohol ASSIST scores at 3 months by 38% versus 21% in the control arm [67]. At 3 months, Noknoy and colleagues found a significant difference in number of binge drinking days between intervention (0.29) and control group (1.36), but at 6 weeks and 6 months, there was no significant difference. At 6 weeks, 3 months, and 6 months, there were significant differences in the average drinks per drinking day between intervention (3.00, 2.73, and 2.26) and the control group (4.85, 5.06, and 4.02), but no difference in the number of drinking days between baseline and follow-up [75]. Nadkarni and colleagues found 36% remission (AUDIT 12–19) of alcohol use in the intervention group compared to the 26% of the control group. At 3 months follow-up, abstinence was significantly higher in the intervention (42%) compared to control (18%) groups. No effect on mean daily alcohol consumption or percent days of heavy drinking differences was found [70]. Results at 12 months showed maintained and enhanced effects on alcohol-related outcomes [71]. A pilot study found that for men with AUDIT>20, the CAP intervention arm had nonsignificant favorable outcomes for remission, proportion of nondrinkers, and ethanol consumption at 3- and 12-month follow-up as compared to enhanced usual care [72]. |
| MI (6) | 1, 3, and 6 months | Alcohol Consumption Questionnaire RAPI score ACRQ APRA Alcohol abstinence ASSIST Drinking days ASI | Segatto and colleagues found significant reduction in alcohol-related problems and alcohol use in the brief intervention and alcohol educational brochure groups but no significant differences between the groups for days of use and amount of use, RAPI, ACRQ and APRA scores, at 3 months follow-up [88]. Signor and colleagues found a significant difference between groups in the reduction of participants consuming alcohol at 6 months follow-up (70% of individuals in the helpline-based brief intervention group and 41% in the control/ minimal intervention group) [89]. Ward and colleagues found that those who received a brief MI at the primary care setting and resource list were more likely to reduce alcohol misuse than control at 3 months [93]. Pal and colleagues found men who received a brief intervention had a decreased average amount of alcohol use in prior 30 days (24.7 to 10.1 versus 26.1 to 19.1) and decreased Addiction Severity Index (0.36 to 0.18 versus 0.42 to 0.33) at 3 months compared to those who received simple advice only [76]. A significant reduction in AUDIT scores at 3 months follow-up was observed by Kamal and colleagues for an on-campus, nurse-delivered brief alcohol screening, and intervention as compared to general advice. The intervention group also had a significant shift of participants from high- to low-risk AUDIT zone as compared to the control group [58]. Shin found no differences in the proportion of abstinent days between intervention and control in a TB clinic [22]. |

*(Continued)*

**Table 2.** (Continued)

| Intervention description | Follow-up time | Outcomes | Summary of findings |
|---|---|---|---|
| BASICS, MI, and harm reduction (1) | 12 and 24 months | # drinks per day RAPI score Harmful alcohol use score (AUDIT) | Simao and colleagues found that college students receiving a brief alcohol screening and intervention had a decrease in the quantity of alcohol use per occasion (4.5 drinks/occasion to 3.7) compared to control (5.1 drinks/occasion to 5.0) at 24 months. There was also significant reduction in AUDIT and RAPI scores between intervention (9.6 to 7.3; 7.0 to 4.3) and control (9.6 to 8.6; 7.6 to 3.9, respectively) [90]. |
| PNF (3) | 1, 3, and 6 months | AUDIT/AUDIT–c Alcohol consequences # of drinks Binge drinking | The intervention group showed a reduction in the number of drinks in a typical drinking day at all follow-up times (OR ranging from 0.71 to 0.68) compared to control. A significant increase in alcohol consequences was observed in the intervention group at 3 months compared to control. The intervention effects were higher for participants with higher motivation for receiving the intervention groups [34]. No differences in binge drinking between control and intervention were observed by Baldin and colleagues [30]. Bedendo and colleagues found in this we-based study of college students a significant reduction in AUDIT scores among NFO and CFO study arms at 1 and 3 months follow-up, respectively, as compared to the PNF arm. Alcohol consequences were lower in NFO at 1 month follow-up and in drinking frequency at 3 months follow-up compared to PNF [35]. |
| Women-focused social cognitive oriented behavioral intervention (1) | 6 and 12 months | Heavy drinking episodes # binge drinking days | Intervention arm showed significantly less frequent heavy drinking behavior (−13.5 in % points) and heavy drinking days (9.9 [SD 8.4] average drinks for control versus 7.4 [SD 7.8] for intervention) at 6 months, but no changes at 12 months. There was no difference in the average number of drinks per drinking days at both follow-up times [94]. |

ACRQ, Alcohol Consumption Risk Questionnaire; APRA, Alcohol Perception of Risk Assessment; ASI, Alcohol Severity Index; ASSIST, Alcohol, Smoking and Substance Involvement Screening Test; AUDIT, Alcohol Use Disorders Identification Test; BASICS, Brief Alcohol Screening and Intervention of College Students; CAP, Counseling for Alcohol Problems; CFO, consequences feedback only; LMIC, low- and middle-income country; MI, motivational intervention; NFO, normative feedback only; OR, odds ratio; PNF, personalized normative feedback; RAPI, Rutgers Alcohol Problem Index; RCT, randomized controlled trial; TB, tuberculosis; WHO, World Health Organization.

long-term (6+ months) outcomes, comparing intervention and control [29,34,35,41, 58,67,70,71,75,76,80,88–90,91,93,94].

WHO-based brief interventions were found to be efficacious to reduce average daily alcohol consumption in males at the health setting [29] and AUDIT average scores in university students [81]. With brief interventions delivered from a motivational intervention framework, Signor and colleagues found at 6-month follow-up, 70% of individuals in the helpline-based brief intervention group and 41% in the control/minimal intervention group had remained abstinent [89]. Similarly, this mode of delivery showed evidence of efficacy at the primary care setting to reduce alcohol use [76,91,93] and with university students [58]. Simao found that college students had a significant reduction in the amount of alcohol use per occasion and AUDIT scores up to 24 months after the intervention [90]. Other modes of delivery revealed that lay counselor–delivered interventions had significant differences between intervention and control [67,71,75], and a computerized intervention reduced alcohol use as much as an in-person motivational intervention [41]. One study focused on the efficacy of a women-focused brief intervention demonstrated efficacy of the interventions in reducing heavy drinking behavior and heavy drinking days in women [94].

However, there were some studies that found a similar reduction in alcohol-related outcomes between both the intervention and control groups, thus a null effect [22,28,30,79,80,82,88]. Assangkornchai and Pengpid found brief interventions at the primary

**Table 3. Meta-summary of studies assessing patient-level interventions to reduce alcohol harms in LMICs: psychotherapy or counseling RCTs.**

| Intervention description | | Follow-up times | Outcomes | Summary of findings |
|---|---|---|---|---|
| CBT (6) | Individual versus group CBT | 15 months | # binge drinking days<br># drinking per drinking days<br>% harmful drinking | Marques and colleagues found that at 15 months, both group and individual interventions had reduction in the mean number of drinking days (group 51 to 29 and 47 to 30), number of heavy drinking days (40 to 20 and 29 to 11), number of problem drinking days (21 to 7 and 12 to 4), mean weekly consumption (43 to 19 and 30 to 12), GGT (109 to 43 and 87 to 34), and SADD (17 to 11 and 17 to 11). There was no difference between the groups [64]. |
| | CBT | 1, 2, and 3 months | SADQ scores<br># drinking days | Satyanarayana and colleagues found that both usual care and CBT for inpatient alcohol-dependent males who screened positive for intimate partner violence reduced SADQ scores over 3 months (ICBI 28.9 to 18.9, 27.3 to 19.7) with no significant between-group differences [87].<br>Papas and colleagues found that compared to usual care, CBT for HIV-infected outpatients who reported hazardous or binge drinking showed a reduction in mean difference percent drinking days (24.9) and drinks per drinking days (2.88) at 30 days follow-up [77]. |
| | CETA, a CBT-based treatment model targeting mental and behavioral comorbidities | 12 months | AUDIT | At 12 months follow-up, Murray and colleagues found a significantly greater reduction in the mean AUDIT score of the CETA intervention arm (14.9 to 5.7) compared to treatment as usual (14.6 to 10.0) in couples with intimate partner violence [68]. |
| | Group CBT versus healthy lifestyle education | 9 months | % drinking days<br>Drinks per drinking day | Papas and colleagues found that compared to healthy lifestyle education, the group CBT intervention arm had significantly lower % drinking days (10.26 versus 7.58) and drinks per drinking day (1.69 versus 1.15) overall [78]. |
| | CBT with CM | 1 month | BrAC | Hartmann and colleagues found that compared to usual care, a significantly greater proportion of individuals receiving CBT with incentive-based CM tested negative for alcohol consumption (0.96 versus 0.76) at 1 month follow-up; incentives-only arm had a similar reduction in alcohol consumption to the CBT with incentive-based CM [53]. |
| Combined methods (5) | Phramongkutklao model, an inpatient rehabilitation program using Buddhism, CBT, health education, family education, and relaxation therapy | 1, 3, and 6 months | Abstinent days<br>Alcohol consumption<br>Craving days | Daengthoen and colleagues found an intensive inpatient rehabilitation model (PMK) found a significant difference in the mean difference of alcohol consumption (mean difference −9.4 baseline, −23.0 1 month, −3.3 3 months, and −4.4 6 months) and mean drink cravings (4.3 versus 3.3) at 1, 3, and 6 months [45]. |
| | Family inclusive relapse prevention | 6 months | % of abstainers days | Nattala and colleagues found a significantly higher percentage of dyadic relapse prevention patients were abstinent throughout the 6-month follow-up period (57%) compared to individual relapse prevention (27%) and treatment as usual (30%) [73]. |
| | MI and PST | 3 months | Harmful alcohol use score (AUDIT) | Sorsdahl and colleagues found for emergency department patients, there was a significant reduction in substance use determined by ASSIST at 3 months for those who received a MI–PST intervention (18.71 to 9.89) compared to the MI (19.96 to 12.28) and control (19.3 to 11.91). There was no significant difference between the MI and control group [92]. |
| | Combined MI and CBT nurse delivered individual counseling | 6 months | AUDIT | At 6-month follow-up, Madhombiro and colleagues found a significantly greater change in AUDIT scores in the intervention arm (14.89 to 8.75) as compared to enhanced usual care (14.74 to 11.61) [63]. |
| | BMS intervention (1), Multidimensional holistic group intervention combining health education and relapse prevention with acupuncture, breathing, and meditation-based exercises | 1, 2, and 3 months | PACS<br>Drinking days<br>Drinks per drinking day<br>Relapse | Ng and colleagues found significantly less alcohol cravings, drinking days, drinks per drinking day, and rates of relapse in the BMS intervention group as compared to treatment as usual at 3-month follow-up [74]. |

(*Continued*)

**Table 3.** (Continued)

| Intervention description | | Follow-up times | Outcomes | Summary of findings |
|---|---|---|---|---|
| MI (4) | MI based counseling sessions, WHO Brief Intervention for Alcohol Use | 6 and 12 months | % of abstinent days # binge drinking days | At 6 months, there were significant reductions in alcohol use over the prior 30 days for the intervention group with 53.8% reporting never drinking over the prior 30 days compared to 26.2% of the control group. Significant reduction in binge drinking with 73.7% of the intervention group compared to 33.2% of the control group reporting never binge drinking in the prior 30 days [61]. At 12 months, 66.3% of the intervention group reported never drinking over the prior 30 days compared to 39.4% of the control group. Similarly, 78.9% of the intervention group reported never binge drinking compared to 47.6% of the control group [61]. |
| | Group-based MI | 3 and 12 months | AEP % harmful alcohol use (risky drinking) | Rendall-Mkosi and colleagues found that compared to the control, a 5-session intervention reduced the proportion of women at risk for AEP (51% intervention and 28% control) at 3 and 12 months. There were declines for both groups in the proportion of women who met criteria for risky drinking at 3 and 12 months (intervention 14.75% versus 10.94%), but the difference between the 2 groups was not significant [83]. |
| | Relapse prevention and MI with or without HVs for outpatients | 3 months | % abstinence Consumption days | Moraes found that after intensive outpatient intervention, of those with subsequent HVs 51.8% were abstinent compared to 43.1% being abstinent among those with no HV controls at 3 months follow-up [23]. |
| | NIH/NIAA-based brief counseling | 6 months | Abstinent days Heavy drinking days | Shin found that for hospitalized TB patients with AUDs who were given a brief counseling intervention with or without naltrexone, there was no change in mean number of abstinent days in the prior 30 days nor number of heavy drinking days [22]. |

AEP, alcohol-exposed pregnancy; AUD, alcohol use disorder; AUDIT, Alcohol Use Disorders Identification Test; BMS, body–mind–spirit; BrAC, breath alcohol concentration; CBT, cognitive behavioral therapy; CETA, Common Elements Treatment Approach; CM, contingency management; GGT, Gamma-Glutamyl Transferase; HV, home visit; ICBI, integrated cognitive-behavioral intervention; LMIC, low- and middle-income country; MI, motivational interviewing; NIAA, National Institute on Alcohol Abuse and Alcoholism; NIH, National Institutes of Health; PACS, Penn Alcohol Craving Scale; PMK, Phramongkutklao; PST, problem solving therapy; RCT, randomized controlled trial; SADD, Short Alcohol Dependence Data Questionnaire; SADQ, Severity of Alcohol Dependence Questionnaire; TB, tuberculosis; WHO, World Health Organization.

care setting addressing alcohol and other substances reduced alcohol use for both intervention and control arms equally [28,80,82]. Similarly, those evaluating a brief intervention in tuberculosis patients in the inpatient or outpatient setting showed limited results [22,79]. Web-based, personalized normative feedback (PNF) interventions showed conflicting results in 2 studies with similar populations [30,34]. Nadkarni and colleagues found a reduction in alcohol consumed and mean AUDIT score for both MI-based interventions but no difference between the groups; however, this feasibility study was not powered to detect such differences [70,72].

**Psychotherapy or counseling.** Overall, 15 RCTs matched our definition of psychotherapy or counseling. Interventions in this group varied in terms of length, population, and framework. Most commonly, interventions used MI techniques [22,23,61,69,72,83] or cognitive behavioral therapy (CBT) [53,64,68,77,78,87]. Some interventions used education and stigma reduction [73] or a combination of methods [45,92]. These studies also had varied populations, including hospitalized patients [45,63,73,74,87,91], emergency department patients [92], outpatient primary care patients [69], and patients visiting clinics specializing in reproductive health [61,83], HIV or tuberculosis [22,77], and substance abuse [23,64].

A number of the studies found a reduction in alcohol-related outcomes. Daengthoen's intensive inpatient rehabilitation combination therapy intervention had a reduction in alcohol consumption and drink cravings [45]. Nattala found that a significantly higher percentage of dyadic intervention patients (57%) were abstinent compared to individual treatment (27%) or treatment as usual (30%) patients [73]. Sorsdahl and colleagues found a reduction in the

**Table 4. Meta-summary of studies assessing patient-level interventions to reduce alcohol harms in LMICs: health promotion and education RCTs.**

| Intervention description | | Follow-up times | Outcomes | Summary of findings |
|---|---|---|---|---|
| Workplace health promotion programs (2) | Team awareness, social cognitive, and MI theory health promotion | 3 months | Binge drinking<br># drinking days<br># drinking per days<br>Attitude toward alcohol | Burnhams and colleagues found that a team awareness intervention reduces the mean binge drinking days from 2.1 to 1.4 days compared to an increase from 1.6 to 2.1 days in the control group [38].<br>Aira and colleagues found at a 3-month follow-up, a health promotion intervention for factory workers had a reduction in alcohol drinks per day for men (b −0.19) and women (b −0.28) and attitudes toward drinking (b 3.06), but not for days of alcohol consumption (b −0.13) [26]. |
| Community based (5) | Community-based intervention for risk reduction of HIV-related behaviors | 12 and 24 months | Current alcohol use<br>Frequency of alcohol use<br>Quantity of drinks consumed | Cubbins and colleagues studied a community-based intervention and found declines in alcohol use and abuse over the study period in relatively equal levels [43]. |
| | HV<br>HV with CM<br>HV for pregnant women | 12, 13, and 16 weeks<br>18 and 36 months | Abstinence<br>Drinking days<br>Alcohol use during pregnancy<br>Frequency of use<br># of drinks per drinking day<br>Frequency of 3 or more drinks per day<br>AUDIT | Moraes and colleagues evaluated the cost-effectiveness of an outpatient conventional (CT) alcohol rehabilitation treatment to conventional treatment with HV. Authors found both groups had a large proportion of the patients were abstinent at follow-up CT (3.4% to 43.1%) and HV (1.6% to 58.11%), but the overall difference of 44% more abstinent patients was not significantly different [23].<br>Jirapramukpitak and colleagues found that home-based CM did not improve continuous abstinence over the 12-week intervention period. The higher-magnitude CM intervention arm did have a significantly higher abstinence rate in the postintervention follow-up period [54].<br>Rotheram-Borus and colleagues studied a HV for prenatal and postnatal visits for pregnant women up to 36 months postdelivery and did not find a direct association between intervention and alcohol use [84].<br>Bolton and colleagues found no differences in AUDIT scores between a control and intervention in Burmese refugees in Thailand [37]. |
| School based (6) | STEP for HIV/AIDS and alcohol use | 10 weeks | Intention to use | Chhabra and colleagues found no differences in intention to use alcohol after implementation of a STEP program compared to control at 10-week outcome assessment [40]. |
| | School-based curriculum using communication competence theory to develop use resistance strategies | 8 months | Drinks per day<br>Drinking days | Marsiglia and colleagues found that after an implementation of a school-based curriculum, both intervention and control groups had an increase in the amount of use and frequency of use, yet the intervention group had significantly less increase in amount and frequency of use [65]. |
| | Socioecological theory and sociocognitive theory–based healthy lifestyle education and environmental changes | 4 months | % participants reporting alcohol intake | Barbosa Filho and colleagues found that no differences between control and intervention groups were observed in the proportion of adolescents reporting not taking alcohol in the last month [33]. |
| | Nurse-delivered Health, Education, Prevention and Self-Care (SEPA) based on Social Cognitive Model of Behavior Change | 1 month | AUDIT | Among sexually active university-recruited women, Mendez-Ruiz and colleagues found decreased alcohol use in the intervention group compared to the control group [66]. |
| | Life skills development curriculum for schools based on a comprehensive social influence program | 9 and 21 months | % of first use of alcohol<br># of binge drinking days<br>% of alcohol use | No differences were observed at 9 months between intervention and control for alcohol use and binge drinking. At 9 months, participants in the intervention group showed a higher chance of using alcohol for the first time (RR 1.30, CI 95% 1.13;1.49) [85].<br>At 21 months, participants in the intervention group reported higher risk of initiating alcohol use (OR 1.13, CI 95% 1.01;1,27) and higher chance of using alcohol in the past year (OR 1.30, CI 95% 1.02;1.65). No effects were observed for binge drinking in the past year or alcohol use and binge drinking in the past month [86]. |

*(Continued)*

**Table 4.** (Continued)

| Intervention description | | Follow-up times | Outcomes | Summary of findings |
|---|---|---|---|---|
| Clinic based (7) | Family Strengthening Intervention for HIV-affected families | 3 months | Caregiver AUDIT | Chaudhury and colleagues found compared to treatment as usual, a family-based intervention to reduce alcohol use and violence within HIV-affected families in Rwanda had had significant reductions in alcohol use compared to control (−0.56) at 3-month follow-up [39]. |
| | HIV–alcohol risk reduction intervention | 1, 3, and 6 months | Alcohol use in sexual context Anticipated outcome of alcohol use | Kalichman and colleagues found that a behavioral risk reduction counseling intervention for sexually transmitted infection clinic patients had a reduction in alcohol use and expectancies that alcohol enhances sexual experiences at 3-month follow-up [57]. Kalichman and colleagues found that compared to a 1-hour HIV–alcohol education group, the 3-hour brief behavioral HIV–alcohol risk reduction intervention reduced alcohol use before sex at 3 and 6 months [56]. Ahmadi and colleagues found that, compared to treatment as usual, an HIV-focused peer education training program had significant reductions in both alcohol use prior to sexual intercourse and number of sex acts while intoxicated among female drug users at 1 and 3 months follow-up [25]. |
| | Group CBT versus healthy lifestyle education | 9 months | % drinking days Drinks per drinking day | Papas and colleagues found that compared to healthy lifestyle education, the group CBT intervention arm had significantly lower % drinking days (10.26 versus 7.58) and drinks per drinking day (1.69 versus 1.15) overall [78]. |
| | HIV SR reduction arm and MI+risk reduction | 3 and 6 months | AUDIT | Witte and colleagues studied the efficacy of a relationship-based SR reduction intervention, SR reduction intervention with MI compared to a wellness control to reduce harmful alcohol use among female sex workers. All groups were effective in reducing the AUDIT score from baseline to 6 months (wellness promotion −30.98 to 18.30, risk reduction −28.42 to 18.12, and risk reduction and MI −32.64 to 21.72), but there was no significant difference between groups [95]. |
| | Multifaceted district level mental healthcare plan + brief intervention | 12 months | AUDIT | Jordans and colleagues found no statistical significant difference between control and intervention for the reduction in AUDIT scores from baseline and follow-up (B = 12.16; CI 95% −6.10; 1.79) [55]. |

AUDIT, Alcohol Use Disorders Identification Test; CM, contingency management; HV, home visit; LMIC, low- and middle-income country; MI, motivational interviewing; OR, odds ratio; RCT, randomized controlled trial; RR, risk ratio; SEPA, Health, Education, Prevention and Self-Care; SR, sexual risk; STEP, School-based Teenage Education Program.

ASSIST scale for those who received the MI with problem solving intervention compared to the control, yet there was no difference between the MI alone and the control group [92]. L'Engle, Rendall-Mkosi, and Moraes all had significant findings for their MI interventions reducing binge drinking up to 12 months, reducing the proportion of women at risk for alcohol-exposed pregnancies and increasing the proportion of abstinent patients [23,61,83]. Ng and colleagues used a body–mind–spirit multidimensional intervention and reported significantly less alcohol cravings, drinking days, drinks per drinking day, and relapse in the intervention group compared to treatment as usual at 3 months [74]. Randomization to receive CBT, in different modalities, was found to be associated with a higher reduction in drinking days, drinks per drinking days [77,78], and AUDIT score [63] in comparison to usual care, at 3 months for HIV-infected outpatients reduction in mean AUDIT score [68], and alcohol consumption [53] in participants positive for intimate violence.

A few of the studies in this subgroup had null effects or found no difference between the intervention and control arms. Marques and colleagues found a reduction in many of their alcohol-related outcomes for the group and individual intervention arms at 15 months, but the

**Table 5. Meta-summary of studies assessing patient-level interventions to reduce alcohol harm in LMICs: biological treatment RCTs.**

| Intervention description | | Follow-up times | Outcomes | Summary of findings |
|---|---|---|---|---|
| Medication (15) | Naltrexone (3) | 1, 2, 3 and months | % abstinent<br>% relapse<br># abstinent days<br># heavy drinking days | There was no significant difference in the percentage of abstinence, number of heavy drinking days, or number of abstinent days when comparing naltrexone and the placebo [22,24,32].<br>Naltrexone significantly decreased the relapse percent when compared with the placebo, 74.14% relapse in control, versus 55.17% relapse in the intervention. [24], although another study reported no significant change in percent abstinent at fourth week or eighth week (intervention: week 4: 53.1% week 8: 40.8% Control: week 4: 42.6% week 8: 31.5%) [32]. |
| | Acamprosate with participation in AA optional (1) | 1, 2, 3, 4, 6, 8, 12, 16, 20, and 24 weeks | % abstinent<br>Abstinent days | There was a significant difference in abstinence between the trial group (acamprosate) (42.5%) and the control group (20%) [31].<br>Abstinent days were significantly greater in patients who received acamprosate and did not participate in AA than in patients who received placebo and did not participate in AA. Abstinent days were not significantly greater in the subgroup who received acamprosate and participated in AA than in the subgroup who received placebo and participated in AA [31]. |
| | Ondansetron (1) | 3 months | % abstinent<br># drinks per day | There was no significant difference between the trial group (ondansetron) and the control group (placebo) for the main outcome, percentage of study participants abstinent (trial: 88.6%, placebo: 76.1%). There was also no significant difference between mean number of drinks per day (trial: 0.66, placebo: 1.09) [42]. |
| | Baclofen + brief intervention (1) | 3 months | # abstinent days | Baclofen and brief intervention (FRAMES) significantly increased the number of abstinent days (65.1) when compared to the benfotiamine (nutritional supplement/control) group and brief intervention (FRAMES) (39.66) [52]. |
| | Gabapentin (1) | 1 month | Drinks per days<br>Drinks per drinking day<br>% heavy drinking<br>% abstinent<br>OCDS | The gabapentin group had a significantly decreased number of drinks per day, weekly drinks, alcohol consumption during 4 weeks of treatment, and mean percentage of heavy drinking days, and a significantly higher mean percentage of days of abstinence. No differences in drinks per drinking day or OCDS scores between groups [51]. |
| | Topiramate (1, 1 repeated sample) | 1, 2, and 3 months | % abstinent<br># drinks per day<br># drinks per drinking day<br>% drinking days<br># heavy drinking days<br># abstinent days | Topiramate caused a significant increase in the percentage of abstinence at 4 weeks compared to the control group, 42.6% in the placebo and 67.3% in intervention [32]. There was no statistical difference between topiramate and the placebo at week 4, week 8, and week 12 for percent heavy drinking days (intervention: 0.7, 4.9, 2.3; control: 5.0, 5.7, 5.3), percent of drinking days (intervention: 6.6, 7.5, 5.5; control: 11.9, 11.3, 6.4), number of drinks per drinking day (intervention: 1.1, 2.9, 1.2; control: 1.7, 2.2, 4.2), and number of drinks per day (intervention: 0.2, 0.7 0.7; control: 0.7, 0.7, 0.9) [62]. |
| | Disulfiram (5) | 9 and 12 months | # of days of abstinence<br># days until relapse<br># drinks per week<br># drinks per occasion<br>OCDS | The groups receiving disulfiram showed higher frequency of days of abstinence, higher days to first relapse, less craving and less relapse events than topiramate in alcohol-dependent men [49].<br>A similar pattern of results were observed when comparing disulfiram to naltrexone, but had higher cravings. No differences were observed in the amount of days to the first alcohol used [46,48,50].<br>Compared to acamprosate, the group receiving disulfiram showed higher abstinent days, fewer relapse events, and a higher number of days until first alcohol use and to first relapse. No difference was observed in the total number of abstinent days and a higher craving was observed in the disulfiram group [47]. |
| | Amitriptyline versus Mirtazapine (1) | 56 days | Alcohol craving | The mean alcohol craving scores decreased significantly from baseline to follow-up in both groups. There were no differences in the craving scores between mirtazapine and amitriptyline groups (170.7 SD 26.0 versus 157.7 SD 29.4 at the baseline and 97.3 SD 40.6 versus 99.9 SD40.2 at the endpoint) [27]. |
| | Escitalopram + electroacupuncture (1) | 4 weeks | PACS | Zhao and colleagues found that after 4-week treatment, the global scores of PACS declined significantly in both the escitalopram with electroacupuncture and the escitalopram without electroacupuncture groups (both $P < 0.05$). Furthermore, the decline in the rea-electroacupuncture group was superior to that in the sham electroacupuncture group ($P < 0.05$) [96]. |
| Brain stimulation (4) | tDCS (4) | Immediate post treatment<br>5 weeks<br>3 and 6 months | Alcohol craving level<br>% of relapse | tDCS significantly decreased alcohol cravings compared to sham stimulation [36,44]. Klauss and colleagues found that the percentage of relapse at 6 months follow-up was higher in the sham group (88%) than the tDCS group (50%) with no difference in cravings between the groups [59]. However, an intensive tDCS scheme was associated with a larger reduction in alcohol cravings when compared to sham-based control, also associated with lower relapse up to 3 months postintervention [60]. |

AA, Alcoholics Anonymous; FRAMES, Feedback, Responsibility, Advice, Menu, Empathy, Self-efficacy; LMIC, low- and middle-income country; OCDS, Obsessive Compulsive Drinking Scale; PACS, Penn Alcohol Craving Scale; RCT, randomized controlled trial; tDCS, transcranial direct current stimulation.

intervention arms were not significantly different from each other [64]. Similarly, Satyanara-yan found a reduction in Severity of Alcohol Dependence Questionnaire (SADQ) scores for CBT and usual care arm patients but no significant difference between intervention arms; authors believed that this was because both intervention arms received similar alcohol reduc-tion strategy intervention components [87]. Alternatively, Shin and colleagues found that their intervention, which focused on inpatient tuberculosis patients with severe AUDs, caused no change in alcohol-related outcomes, likely because the study did not include alcohol treat-ment–seeking patients, but had patients with low readiness to change or poor intervention participation rates combined with relatively low enrollment numbers [22].

**Health promotion and education.** In total, we found 20 RCTs, which evaluated health promotion and education interventions. Of these, 2 were based in the workplace [26,38], 5 in the community [23,37,43,84], 6 in schools [33,40,65,85,86], and 7 in clinics [39,55–57], of which 1 was focused on women sex workers [95]. The majority of programs addressed alcohol use in the context of HIV/AIDS prevention and risk reduction [38–40,43,56,57,95].

About half (8 of 17) of the health promotion and education interventions were found to have positive results [25,26,38,39,54,56,57,65,66,84]. Some of these studies also had mixed results. For example, Aira and colleagues found a reduction in drinks per day and an improve-ment in attitudes toward drinking, but not a reduction in the total amount of alcohol con-sumption [26]. Similarly, Rotheram-Borus found that home visits for pre- and postnatal women were associated with a reduction in the use of alcohol during pregnancy, but this drinking resumed postpartum [84].

Meanwhile, a majority of the studies that found no effect of their interventions either were not adequately powered to detect the alcohol-related outcome [23,43] or were compared to another intervention rather than a control, thus potentially obscuring some potential reduc-tion in harm [95]. Cubbins and colleagues evaluated a community-level intervention in which popular community individuals relayed education through casual conversations and found significant alcohol reduction in both the intervention and control groups, but no difference between the groups [43]. Chhabra and colleagues looked at the effectiveness of a Severity of Alcohol Dependence Questionnaire (STEP) school-based program but found that students, and more specifically girls, had an immediate reduction in their intent to use alcohol, but there was no difference in the intention to use alcohol at the 10-week outcome assessment [40].

**Biomedical treatments.** The final group of RCTs evaluated biomedical treatments and included 19 RCTs evaluating medications and brain stimulation. The 14 RCTs evaluating medications looked at naltrexone (3) [22,24,32], ondansetron (1) [43], gabapentin (1) [51], disulfiram (5) [46–50], and topiramate (2) [32,62]. Two RCTs evaluated combined behavioral and medication interventions: evaluated acamprosate with Alcoholics Anonymous (AA) [31] and one evaluated baclofen with a brief intervention [52]. One RCT evaluated the efficacy of adding acupuncture to an escitalopram treatment regimen [96].

The RCTs evaluating naltrexone and ondansetron found limited impact on abstinence, number of heavy drinking days or number of abstinent days [22,24,32], and abstinence [43]. Mixed effects were found by RCTs for topiramate, where Baltieri found an increase in absti-nence at 4 weeks, although Likhitsathian found no differences in any drinking quantity or fre-quency measures up to 12 weeks [32,62].

On the other hand, the RCTs evaluating gabapentin, acamprosate, and baclofen exhibit more positive results. Furieri found that gabapentin was associated with a significant decrease in quantity and frequency of drinking and higher mean abstinent days [51]. Baltieri found that acamprosate improved abstinence rates but only for those who participated in AA [31]. More-over, Gupta found that patients who received baclofen compared to a nutritional supplement,

with a brief motivational intervention, were more likely to remain abstinent, have lower heavy drinking days, and fewer alcohol cravings [52].

We identified 4 RCTs that studied transcranial direct current stimulation, and all of them occurred at 2 institutions in Brazil. While 3 of these studies found a decrease in alcohol cravings compared to sham stimulation [36,44,60], one study found a lower relapse rate in the brain stimulation group but with no difference in alcohol cravings at 6-month follow-up [59]. Ultimately, 3 of the 4 studies in this group found more relapses in the intervention group at 4-week, 6-month, and 12-month follow-up [44,59,60].

## Discussion

This is the first review, to our knowledge, of alcohol harm reduction interventions evaluated in LMICs. Most studies we found took place in middle-income countries; there was a noticeable gap of studies in the Middle East, North Africa, Europe, Central Asia, and South Asia regions. Overall, we found that there was limited uniformity for interventions, outcomes, and follow-up times across studies, which limited our ability to compare results. The vast majority of evaluations were limited to middle-income settings, leading to feasibility and generalizability concerns for low-income settings. Of all the RCTs, brief interventions were the most commonly studied; similarly, MI techniques were the most prevalent behavior change technique common in both brief and psychotherapy and counseling interventions. Brief interventions and motivational interviewing techniques also had the most consistent positive results in our findings.

### Lack of uniformity limits effective comparisons

The studies included in our meta-summary used a wide variety of metrics to measure alcohol-related outcomes of alcohol interventions; these metrics included (i) AUDIT scores; (ii) ASSIST scores; (iii) # of drinking days; (iv) # heavy drinking days; (v) # drinks per drinking day; (vi) # abstinent days; (vii) # drinks per day; (viii) % of patients abstinent; and (ix) % of patients relapsing.

The time period over which these outcomes were measured also varied considerably, from 3 months [41] to 24 months [90]. This lack of uniformity compromised our ability to discern the effectiveness of interventions or to compare results across studies. The diversity of alcohol consumption outcomes measures is due in part to varying recall, reference period, and definition of a "standard drink" [97–99]. Future study studies may benefit from using consistent outcome measures and adopting uniform methods of intervention implementation or study designs.

### Uncertain feasibility of implementing interventions in low-income country setting

The vast majority of the studies in this review were conducted in middle-income countries. Thus, the feasibility of implementing these interventions and their effectiveness in low-income settings is uncertain. Low-income countries face greater barriers (such as scarcity of medical facilities, limited training available to medical staff, infrastructural barriers to healthcare access, and effective patient communication/follow-up) to implementing effective healthcare than either high- or middle-income countries. As a clear example, all 4 studies [36,44,59,60] that used brain stimulation as an intervention were conducted in Brazil, an upper middle-income country. In addition to its uncertain effectiveness, brain stimulation requires expensive equipment and specific facilities, and it is not likely to be feasible in some low-income country settings. In another example, although psychotherapy and counseling interventions are demonstrably effective [45,61,73,83,92], none of these studies took place in a low-income

country, so the feasibility of implementing this type of intervention is uncertain. Given that infrastructure in low-income settings is even more limited in mental health and substance abuse facilities and professionals, with a greater associated stigma, an alcohol use reduction intervention implementation of this kind is still potentially unfeasible. Similarly, medication shows some evidence of a positive effect on abstinence from alcohol, but reliable availability of medication is essential for this intervention to be effective; thus, medication may not be a feasible intervention in a low-income country [100,101].

Instead, the most studied and potentially most feasible intervention is a brief intervention. In our systematic review, 6 studies evaluated brief interventions in South Africa [67,79–81,93,94]. Brief interventions have been studied to decrease alcohol use and alcohol-related consequences in a variety of settings and countries [102–105]. They have also been suggested to be cost-effective in high-income countries [106]. In low- and middle-income settings, brief interventions are likely to be feasible because they can be delivered by nonprofessionals requiring less training.

## Limitations

There are a few limitations to our study. First, our search strategy did not exclude studies due to language, and, yet, we found no manuscripts in other languages. Thus, either there is no non-English language literature on this topic or the data sets we searched have limited non-English language articles. Second, our ability to conduct a thorough meta-analysis was restricted by nonuniform outcome measurements, a wide variety of outcome assessment times, and a wide variety of interventions, making it difficult to compare interventions and their effects. To compensate for this, we summarized results from RCTs qualitatively. Similarly, we conducted our database search for only LMICs at that time. This might limit our findings by excluding articles from countries that have become high income since the study occurred or erroneously including countries that were high income but then reduced their status at the time of the database search; in the former case, we cannot determine the number of potential studies, but for the latter case, we rechecked the World Bank status of all countries and their intervention time periods to ensure this was not occurring. Finally, we tried to group types of interventions based on standard definitions rather than study-specific descriptions that might limit the interpretation of effect size and differ from the original author's description.

## Improving future research

Future alcohol harm reduction intervention studies should use uniform reporting. Studies ranged widely in their intervention type, framework, population, augmentation, or boosters, as well as outcome assessment frequency and timing. Adherence to 1 or 2 sets of standardized outcome reporting measures at a specified time period would greatly improve comparability across time and geographic location, allowing for a meta-analysis of intervention methods. Based on our review, brief interventions using the ASSIST or AUDIT scoring systems are the most widely used and appear to provide the best standardization among outcomes. Overall, future research should include both comparative effectiveness to determine best interventions for LMIC settings but also most effective implementation strategies including target populations.

## Conclusions

In conclusion, alcohol harm reduction interventions in LMICs are nonuniform in nature, skewed in geographic regions where applied, and result in uncertain effectiveness over varying

time horizons. Feasible options specific to low-income countries are most likely brief interventions and interventions that utilize motivational interviewing techniques. Identifying uniform methods of implementation and assessment of alcohol harm reduction interventions can be a first step toward establishing a set of evidence-based protocols for treatment for low-income settings. Current studies in brief interventions, psychotherapy, and brain stimulation show promise, but have been tested primarily or exclusively in middle-income settings. Feasibility testing in low-income settings, comparative effectiveness testing, and uniform reporting methods are needed to help determine the most effective alcohol harm reduction strategies for low-income settings in order to address this global health crisis.

## Supporting information

**S1 Fig. Search strategy.**
(DOCX)

**S2 Fig. Map of study location and intervention type.** Source: Global Administrative Areas (2022). University of California, Berkely. Available online: http://www.gadm.org [11/03/2022]; https://geodata.ucdavis.edu/gadm/gadm4.0/gadm404-shp.zip.
(TIF)

**S1 Table. PRISMA Checklist.** PRISMA, Preferred Reporting Items for Systematic Reviews and Meta-Analyses.
(DOCX)

## Author Contributions

**Conceptualization:** Catherine A. Staton, João Ricardo Nickenig Vissoci, Charles J. Gerardo.

**Data curation:** Catherine A. Staton, João Ricardo Nickenig Vissoci, Deena El-Gabri, Konyinsope Adewumi, Tessa Concepcion, Shannon A. Elliott, Daniel R. Evans, Sophie W. Galson, Charles T. Pate, Lindy M. Reynolds, Nadine A. Sanchez, Alexandra E. Sutton, Charlotte Yuan, Alena Pauley, Megan Von Isenberg.

**Formal analysis:** Catherine A. Staton, João Ricardo Nickenig Vissoci, Charlotte Yuan, Alena Pauley, Jinny J. Ye.

**Methodology:** Catherine A. Staton, João Ricardo Nickenig Vissoci.

**Supervision:** Catherine A. Staton, João Ricardo Nickenig Vissoci, Charles J. Gerardo.

**Visualization:** João Ricardo Nickenig Vissoci, Luciano Andrade.

**Writing – original draft:** Deena El-Gabri, Konyinsope Adewumi, Tessa Concepcion, Shannon A. Elliott, Daniel R. Evans, Charles T. Pate, Lindy M. Reynolds, Nadine A. Sanchez, Alexandra E. Sutton.

**Writing – review & editing:** Catherine A. Staton, João Ricardo Nickenig Vissoci, Charlotte Yuan, Alena Pauley, Jinny J. Ye, Charles J. Gerardo.

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
