## [Decision Letter · Decision Letter 0]

30 Oct 2019

Dear Dr. Staton,

Thank you very much for submitting your manuscript "Interventions to Reduce Alcohol-Related Harm in Low- and Middle-Income Countries: a Systematic Review and Meta-Summary" (PMEDICINE-D-19-02160) for consideration at PLOS Medicine for our upcoming special issue on substance mis/use. 

Your paper was discussed among the editorial team and sent to independent reviewers, including a statistical reviewer. The reviews are appended at the bottom of this email and any accompanying reviewer attachments can be seen via the link below:

[LINK]

In light of these reviews, we will not be able to accept the manuscript for publication in the journal in its current form, but we would like to invite you to submit a revised version that fully addresses the reviewers' and editors' comments. You will appreciate that we cannot make a decision about publication until we have seen the revised manuscript and your response, and we expect to seek re-review by one or more of the reviewers. 

We hope to receive your revised manuscript within two weeks. Please email us (plosmedicine@plos.org) if you have any questions or concerns.

Please let me know if you have any questions. Otherwise, we look forward to receiving your revised manuscript soon. 

Sincerely,

Richard Turner PhD, for Philippa Berman, MBBS

rturner@plos.org

We ask you to update your search to a date in the last few months. 

Please restructure the abstract of your paper so that there are three sections: background, methods and findings, and conclusion. The final sentence of the "methods and findings" subsection should summarize the study's main limitations. 

Please add a sentence, say, to your abstract detailing the interventions that have showed promise in randomized controlled evaluations.

After your abstract, we will need to ask you to add a new and accessible "author summary" section in non-identical prose. You may find it helpful to consult one or two recent research papers published in PLOS Medicine to get a sense of the preferred style. 

Throughout your text, please ensure that reference call-outs fall before punctuation, e.g. "... follow-up [42].".

Where you make a claim of "the first", e.g. at line 236, please add "to our knowledge" or similar. 

In the reference list, please ensure that journal names are abbreviated as appropriate, e.g. "Lancet" rather than "the lancet". 

Please complete references 50 & 78, and provide additional information for reference 14 & 21 (including URLs if available, along with accessed dates).

In figure 1, two boxes contain "(n = )" which we ask you to complete as appropriate. 

Please adapt the attached PRISMA checklist so that individual items are referred to by section (e.g., "Methods") and paragraph number rather than by line or page numbers, as the latter generally change in the event of publication. 

Comments from the reviewers:

*** Reviewer #1: 

I confine my remarks to statistical aspects of this paper. The general approach is fine, but I have some issues to resolve before I can recommend publication.

Line 45 - No meta-analysis appears to have been done. I think this is the right choice, but it makes this line incorrect. 

Line 62 - I don't think "standardize" is the right word here. I am not sure what the authors mean. Maybe it is the interventions that need to be standardized? 

Figure 1 - there are two blanks (e.g. records after duplicates removed). Also, the box for articles excluded that were not RCT should come out of "full text articles assessed". Lastly, I think the bottom box should either get N = 0 or be dropped. 

Peter Flom

*** Reviewer #2: 

1) Sorry, but the title and search terms would suggest that all interventions were taken into consideration. This would include the "best buys" (taxation increases, ban on marketing, limitations of availability). Since these interventions have been designated by WHO and others as the most appropriate also in LMIC, I find it curious that they were left out without good reasoning.

2) Where was it stated that the final studies had to be RCTs? This has to be made more explicit.

3) Title and descriptions have to be changed to correspond to the limitations of the actual design.

4) Reference list at times sloppy without page numbers. Also, of course, the searches should have been wider with different data bases, if in the end only individual-based interventions were selected.

5) Minor point: status of WB classification should be mid-term of the interventions and not date of publication.

Final point: while the authors are technically right to indicate the different measurement of outcome, the variance for ALL measures is de facto determined by QF by > 60% -> this means Z-standardization and meta-analysis would be possible.

*** Reviewer #3: 

Comments: Minor Revision

This paper represents a carefully done systematic review of mostly clinical studies for alcohol treatment. The introduction makes a compelling case for the importance of reviewing what is available of effective individual-level interventions, given the importance of reducing hazardous drinking. This could be a great compendium to be used as a resource. There are however some important areas that need to be addressed to make a significant contribution. 

First, the abstract and introduction should not have categorized the study as a meta-analysis but rather as a systematic review. Second, the title should clarify that this systematic review is not of all types of interventions as stated in the introduction ("review and describe the current published literature on alcohol interventions in LMICs") but rather limited to a certain group of interventions. The Discussion section should clarify that many other types of interventions are excluded (i.e. policy, social marketing) from the review, including population-based interventions. Third, the criteria and words used for the search do not explicitly say that they had to be randomized trials, or the wording to only include brief interventions, health education, medication, psychotherapy, or brain stimulation studies. How did the authors decide on these types of interventions? Nor do the authors state why other types of studies were excluded. The abstract says all languages were included but state that "we found no manuscripts in other languages." Could it be due to the data sources included? Given the large body of studies in South America, this seem as a surprise. Fourth, Figure 1 needs to be revised because there is missing information (n=?).

Other concerns deal with the constant shift describing settings (where the interventions were done), type of population served by the interventions, without acknowledging whether the content of the alcohol intervention component of the interventions differed. At other times, when describing the studies, more detail was provided of the content but again there was a constant shift in what of the interventions was being described (i.e. setting, population, content, type of evaluation, format of the intervention, results of the intervention, country where the intervention was conducted, time of the assessment). This made following the results of the systematic review extremely cumbersome. A more consistent approach to describing the studies is necessary to help guide the reader.

The idea that a nurse can deliver the intervention and "even" be effective seems to undermine the value of nurses. Can the authors justify? The writing in some sections also seems choppy (lines 203-205, 218 and others) and sometimes unclear. For example, what do the authors mean by: "psychotherapy or counseling most frequently utilized a blend of theoretical bases"; "limited standardization for outcomes across studies"; "wide variability in the metric of improvement chosen"; "standardization of outcome measures as well as designing interventions"; or "augmentation with additional types of treatment"? Also unclear was why in some sections the authors of the interventions are mentioned but not in other sections, seemingly as if different sections of the paper were written by different authors. 

The Discussion section should specify why the authors focused on alcohol harm reduction interventions, which is only stated at the end of the synthesis but not mentioned as a term in the search for topics.

***

[LINK]

---

## [Decision Letter · Decision Letter 1]

20 Dec 2020

Dear Dr. Staton,

Thank you very much for submitting your revised manuscript "Interventions to Reduce Alcohol-Related Harm in Low- and Middle-Income Countries: a Systematic Review and Meta-Summary" (PMEDICINE-D-19-02160R1) for consideration at PLOS Medicine. We do apologize for the long delay in sending you a decision. 

Your paper was evaluated by the editors and re-seen by our reviewers, including a statistical reviewer. The reviews are appended at the bottom of this email and any accompanying reviewer attachments can be seen via the link below:

[LINK]

Considering these reviews, we will not be able to accept the manuscript for publication in the journal in its current form, but we would like to invite you to submit a further revised version that fully addresses the reviewers' and editors' comments. You will recognize that we cannot make a decision about publication until we have seen the revised manuscript and your response, and we may seek re-review by one or more of the reviewers. 

We hope to receive your revised manuscript after the holidays. Please email us (plosmedicine@plos.org) if you have any questions or concerns.

Please let me know if you have any questions. Otherwise, we look forward to receiving your revised manuscript soon. 

Sincerely,

Richard Turner, PhD

rturner@plos.org

Please finalize the arrangements for data deposition.

Please update the search to a date in the last 3 months. 

Please quote the date of the search in the abstract. Also, please quote the proportions of studies carried out in the different country categories - low-income countries, etc. 

We suggest adding "people in" at line 37. 

Please revisit the phrasing at line 61, where the second "that" should perhaps be removed. 

It seems that motivational interviewing interventions fall into both the "brief interventions" and "psychotherapy" categories. Please detail the key differences underlying these categorizations. 

At line 128, please refer to the attached checklist (e.g., "See S1_PRISMA_Checklist") and rename the attachment to match. 

Throughout the text, please remove spaces from within the reference call-outs, e.g., " control arms equally [26,71].".

Please remove the information on financial disclosures from the end of the main text. In the event of publication, this information will appear in the article metadata via information provided in the submission form. 

In the reference list, please use journal name abbreviations, e.g., "PLoS Med.", consistently. 

Please spell out fully the institutional author name for reference 3 and any other relevant references.

Please ensure that all references contain full access details, e.g., reference 9.

We did not find supplementary table 1 with the paper - please ensure that this is provided with your resubmission. 

Comments from the reviewers:

*** Reviewer #1: 

The authors have addressed my concerns and I now recommend publication

Peter Flom

*** Reviewer #2: 

While the review improved substantially, I still see problems with this revised version (Referring to the version marked):

1) The new title is: "Patient-level Interventions to Reduce Alcohol-Related Harm ...

Sorry but brief interventions by definition are not directed at patients -> see WHO manual on brief interventions.

2) The interventions are now dichotomized at 'brief interventions' and biological treatment. Most treatment interventions are not biological but talk therapy.

3) This reviewer is still concerned with the lack of a meta-analysis, which would be possible.

4) The references continue to be sloppy and old. I just consider the first 6 with comments as examples:

1. Lim SS, Vos T, Flaxman AD, Danaei G, Shibuya K, Adair-Rohani H, et al. A comparative risk assessment of burden of disease and injury attributable to 67 risk factors and risk factor clusters in 21 

regions, 1990-2010: a systematic analysis for the Global Burden of Disease Study 2010. The Lancet. 2012;380(9859):2224-60.

This is an arbitrary citation. There have been several updates, currently to GBD 2017, and next week to GBD 2018 (all in Lancet).

2. Rehm J, Samokhvalov AV, Neuman MG, Room R, Parry C, Lonnroth K, et al. The association between alcohol use, alcohol use disorders and tuberculosis (TB). A systematic review. BMC Public Health. 2009;9:450.

There would be newer overviews by Imtiaz and colleagues.

3. Organization WH, Unit WHOMoSA. Global status report on alcohol and health, 2014: World Health Organization; 2014.

This citation two problems: the treatment of WH in WHO as names, and again, the last Global status report is from 2018.

4. WHO Expert Committee on Problems Related to Alcohol Consumption. Second report. World Health Organ Tech Rep Ser. 2007(944):1-53, 5-7, back cover.

Not sure, what the 1-53, 5-7, back cover should indicate to the reader.

5. Moss HB. The impact of alcohol on society: a brief overview. Social work in public health. 2013;28(3-4):175-7.

6. Chisholm D, Rehm J, Van Ommeren M, Monteiro M. Reducing the global burden of hazardous alcohol use: a comparative cost-effectiveness analysis. Journal of studies on alcohol. 2004;65(6):782-93.

This reference has been updated with important examples from low- and middle income countries on the generalized cost-effectiveness of the interventions dealt with in the publication:

Chisholm, D., Moro, D., Bertram, M., Pretorius, C., Gmel, G., Shield, K., & Rehm, J. (2018). Are the "best buys" for alcohol control still valid? An update on the comparative cost-effectiveness of alcohol control strategies at the global level. Journal of Studies on Alcohol and Drugs, 79(4), 514-522. doi:10.15288/jsad.2018.79.514 

*** Reviewer #3: 

Comments of Reviewer #3:

This papers has been drastically improved and can be informative of the literature available on patient-level alcohol-related interventions in low and middle income countries. I do have a series of comments that require final attention.

Please pass speller to ensure that there are no typographical errors such as "analyses".

The abstract and highlights of the study indicate that the most consistently positive results were brief interventions. "Brief interventions" does not qualify what types of interventions or define it. There is a plethora of brief interventions so the reader needs to have a more concrete picture to what types of interventions. Please use the definition in lines 198-200 to specify what you mean from the beginning of the manuscript.

Given the absence of conclusive evidence of effective alcohol interventions in LMIC, is it adequate to recommend "further evaluations should focus on the most effective implementation strategies in low- and middle-income settings?" Or should it be on testing the comparative effectiveness of these brief interventions in different LMIC contexts to determine what works for whom and under what conditions? This is particularly important since the authors state that most studies are "limited to middle-income settings" and that "lack of uniformity compromised our ability to discern the effectiveness of interventions."

There is some confusion in stating that meta-analyses and systematic reviews were excluded, and then saying that there were no systematic reviews or meta-analysis in LMIC. Which of the two is it? It cannot be both.

The statement in line 142 "Studies which had the same population represented had the most recent data included in the review" is not clear. Please clarify if what you mean is that if two studies used the same data, the most recent study was selected for representation.

Please clarify what you mean by "No studies were excluded for language" bit then authors state "the datasets we have searched are not extensive enough to include non-English language articles."

In lines 304-305, it is unclear if the arms are intervention arms and the comparison is across different treatments or across a treatment and a control arm. Please specify.

Finally, some sentences need to be rephrase for clarity: "Given infrastructure in low-income settings

410 has even more limited mental health and substance abuse facilities and professionals, and

411 greater associated stigma; implementation of this kind of intervention is still potentially

412 unfeasible."

Overall, this manuscript has great potential to be of interest to readers of the journal but requires further attention.

***

[LINK]

---

## [Editor Report · Decision Letter 2]

22 Feb 2022

Dear Dr. Staton,

Thank you very much for re-submitting your manuscript "Patient-level Interventions to Reduce Alcohol-Related Harms in Low- and Middle-Income Countries: a Systematic Review and Meta-Summary" (PMEDICINE-D-19-02160R2) for consideration at PLOS Medicine. We do apologize for the long delay in sending you a response. 

We have discussed the paper with our academic editor and I am pleased to tell you that, provided the remaining editorial and production issues are fully dealt with, we expect to be able to accept the paper for publication in the journal.

[LINK]

Please let me know if you have any questions, and we look forward to receiving the revised manuscript.   

Sincerely,

Richard Turner, PhD

rturner@plos.org

Requests from Editors:

Please remove the information on data availability from the end of the main text: this should be detailed in the submission form (to appear in the article metadata in the event of publication). We were unable to open the specified file, and it may be that some additional information is needed. 

In line 2 of the abstract, please adapt the text to "... have been shown to reduce alcohol use" or similar. 

In the abstract, please adapt the sentence summarizing study numbers to "These RCTs evaluated ...". 

We feel that a few words need to be added to the abstract to explain the nature of "brief interventions". One way to achieve this would be to add a new sentence immediately after that beginning "These RCTs evaluated ..." in which you give 1-2 examples of each category of intervention (e.g., "Brief interventions included ..."). 

Please adapt the text "Due to high heterogeneity of intervention types ..." in the abstract to state that a meta-analysis was not carried out for this reason (removing that point from the final sentence of the "Methods and findings" subsection). 

In the first point of the Author summary, please make that "high rates" or similar (to avoid the implied comparison of "higher"). 

In the second point, please adapt the wording of the second point to "In order to investigate the potential for a patient-level intervention ..." or similar. 

At the end of the Introduction, please move the PICOS sentence to early in the Methods section. 

In the methods section, under "Study selection", it may be helpful to add a few words to clarify the meaning of "six pairs of reviewers" (followed by 11 sets of initials), e.g., "from" the specified individuals; similarly, the "third reviewer" might be "DG or CS". 

In the results section, under "Brief interventions", please avoid "wasn't".

In the results section, under "Biomedical treatments", please correct the apparently broken reference (Correa, 2013). 

Noting reference 12, please remove the information on version from the citation. 

Noting reference 15, please remove all iterations of "[Internet]" from the reference list.

In reference 29, please reverse the current order of study group name and title. 

We suggest adapting the title of table 1 to "All randomized controlled studies (75)".

Please convert figure 2 to a supplementary figure. 

Please confirm that the map in figure 2 can be published under a CC-BY licence (in principle permitting commercial re-use). 

***

---

## [Editor Report · Decision Letter 3]

9 Mar 2022

Dear Dr Staton, 

On behalf of my colleagues and the Academic Editor, Dr Alegria, I am pleased to inform you that we have agreed to publish your manuscript "Patient-level Interventions to Reduce Alcohol-Related Harms in Low- and Middle-Income Countries: a Systematic Review and Meta-Summary" (PMEDICINE-D-19-02160R3) in PLOS Medicine.

PRESS

Sincerely, 

Richard Turner, PhD 

rturner@plos.org